# Time evolution of the behaviour of Brazilian legislative Representatives using a complex network approach

**Ludwing Marenco**[1]*, **Humberto A. Carmona**[1], **Felipe Maciel Cardoso**[2], **José S. Andrade Jr.**[1], **Carlos Lenz Cesar**[1,3,4]

**1** Departamento de Física, Universidade Federal do Ceará, Fortaleza, Ceará, Brasil, **2** Institute for Biocomputation and Physics of Complex Systems (BIFI), University of Zaragoza, Zaragoza, Spain, **3** Institute of Physics "Gleb Wataghin", State University of Campinas, Campinas, São Paulo, Brazil, **4** National Institute of Science and Technology on Photonics Applied to Cell Biology (INFABIC), Campinas, São Paulo, Brazil

* ludving@fisica.ufc.br

**Data Availability Statement:** The data underlying the results of this study are third party. The data containing the roll-call vote sequences of Representatives that support the findings of this

## Abstract

The follow up of Representative behavior after elections is imperative for a democratic Representative system, at the very least to punish betrayal with no re-election. Our goal was to show how to follow Representatives' and how to show behavior in real situations and observe trends in political crises including the onset of game changing political instabilities. We used correlation and correlation distance matrices of Brazilian Representative votes during four presidential terms. Re-ordering these matrices with Minimal Spanning Trees displays the dynamical formation of clusters for the sixteen year period, which includes one Presidential impeachment. The reordered matrices, colored by correlation strength and by the parties clearly show the origin of observed clusters and their evolution over time. When large clusters provide government support cluster breaks, political instability arises, which could lead to an impeachment, a trend we observed three years before the Brazilian President was impeached. We believe this method could be applied to foresee other political storms.

## Introduction

Democracy nowadays faces several challenges, especially for those concerned with the rules adopted to choose Representatives and, subsequently, to follow their behavior in office. Modern society must deal with all sorts of threats, at national and global levels, which range from security to nature preservation, and social impacts of scientific and technological changes. Democratic societies do not delegate the decisions concerning a regulation, or response to a threat, to a team of experts, but rather to Representatives capable of listening and answering to community thinking and needs. However, once elected, no Representative can completely perceive all the issues related to their actions in their full social impact range, meaning that they need to be in closer contact with those they represent and continuously receive their feedback. As a consequence today's democracies do not have to worry about only on elections free of

study are available in http://www2.camara.leg.br/. As required for the Brazilian transparency laws, it is possible to access all information associated to legislative activities. In order to obtain the results of roll-call vote of Brazilian Chamber of Deputies at the website you must go through the navigation menu to "Atividade Legislativa" and select "Plenário" option and finally select "Resultado das votações e lista de presença". The data can be downloaded as DBF or TXT files. The authors used the 52, 53, 54, and 55 legislature TXT datasets. They did not have special access privileges.

**Funding:** We declare that all funding sources are provided by Conselho Nacional de Desenvolvimento Científico e Tecnológico (CNPq), Fundação de Amparo à Pesquisa do Estado de São Paulo (FAPESP), Coordenação de Aperfeiçoamento de Pessoal de Nível Superior (CAPES), Fundação Cearense de Apoio ao Desenvolvimento Científico e Tecnológico (FUNCAP), and the National Institute of Science and Technology for Complex Systems in Brazil. The funders had no role in study design, data collection and analysis, decision to publish, or preparation of the manuscript.

**Competing interests:** The authors have declared that no competing interests exist.

fraud, but also about information concerning their Representative choices. For example, our analysis shows the emergency of consensus for each legislature, which raises the question: are these consensuses in agreement with voter's interests, or do they represent representative betrayal?

Given the importance of the legal system in daily life, it is not surprising that complex network formalism has been extensively adopted to describe intrinsic features of legislative systems. In the last few years, the science of complex systems [1–3] appeared to describe complex networks [4–6] ascribing a measurement of the relationship between network members. The first task for this kind of analysis, therefore, is to define and measure relationship from the data, followed by the use of available graph theory methods [7] in order to characterize the resulting network to extract repetitive patterns and draw objective conclusions. Network science has been able to analyze co-evolution of marginalization in social transition [8]; describe the heterogeneity and interaction of human activity in social networks [9, 10]; explain how inheritance defines the structure of animal networks [11]; and showed the influence of precipitation rates on community structures [12, 13]. Besides, the network science offers general outlines for functional brain networks [14–17], market dynamic analysis [18–21] and others [22–29].

In the political area, the rise of partisanship in the U.S House of Representatives was studied by counting the number of yea/nay in roll-call votes, a measure of the degree of the ideological relationships among members of different parties. The networks constructed using this approach revealed that partisanship and non-cooperation had been increasing exponentially over the last 60 years, producing negative effects over legislative productivity [30]. Complex networks have also been utilized to analyze legislative co-sponsorship in the U.S House and Senate [31, 32]. The co-sponsorship network is typically denser than the legislative one, demonstrating the influence of institutional arrangements and strategic incentives on co-sponsorship. Moreover, this type of network can be used to identify influential legislators [31] and to correlate the community structure of the legislative system with political ideologies [32]. From the roll-call vote sequences, it was possible to create a bipartite network among the committees and Representatives in the U.S House [33, 34], disclosing an inner hierarchical structure of the U.S. legislative system. Its political and organizational features could then be characterized without the need to introduce political information. Under the same framework, another study [35] focused on party polarization by using modularity measures over the set of roll-call votes, and the existence of strategies adopted by elites to preserve the political order could be detected. Finally, the small-world property observed for legislative networks [36] seems to indicate that co-sponsorship is a form of communication which increases the effectiveness of the Congress.

We used complex system techniques to observe the Brazilian Federal Chamber's behavior for a period of 16 years (4 legislatures) which includes one Presidential impeachment. We observed a characteristic pattern in most legislatures that was not present in the tumultuous years around the impeachment. Therefore, our methodology shows signs of political instability in a country. Although some might consider this to be a case specific to Brazil we argue that the same pattern could be found in countries with political instabilities. We used an unbiased framework, based on the network of correlations among roll-call sequences of votes of Representatives, which do not depend on the content of each voted bill or previous knowledge of partisan relationships. Therefore, the patterns observed emerged by themselves, highlighting the regularities and anomalies.

Our methodology did not involve asking questions of the Representatives about their choices in hypothetical situations, but rather to observe their real public choices. This way we avoided the always present bias in the selection/phrasing of questions brought to respondents.

Besides the questioning bias, the response to a real situation can be quite different from the answer to a hypothetical situation. In economy, the consumer preference is revealed only after the purchase choice is made. Here we only observed Deputy's real votes, not answers to hypothetical questions. The politicians are influenced not only by their own beliefs and their electorate, but also by their party, fellows, funding groups, government actions, negotiations among their peers, and so on. Therefore, our analysis of the **REVEALED** politician preferences shall contain not only information about politician/electorate beliefs and thinking, but also on politician's response to surrounding interactions. There was no issue of privacy in our analysis because our data came from the public available records of the votes required by Brazilian transparency laws.

## Results

The Brazilian Federal Chamber is composed of 513 Deputies elected in general elections, together with the President and State Governors, to legislate during a four-year term. Our work includes most general states for the stance of Representatives such as obstructionism, abstentionism and absenteeism, instead of just yea/nay, because these other states also provide important information. In order to vote at the plenary sessions, each Deputy has six options for expressing his stance in favor or against the bill under discussion. Here we quantify these votes in terms of the following random variable

$$V = \begin{cases} 2 & \text{if Yea,} \\ 1 & \text{if Abstention,} \\ 0 & \text{if Art. 17, Present,} \\ -1 & \text{if Absence,} \\ -2 & \text{if Obstruction,} \\ -3 & \text{if Nay,} \end{cases} \tag{1}$$

where the negative values indicate the tendency to refuse, while the positive values express the intention to approve the bill voted. "Art. 17" is an article that prevents the speaker of the Chamber to vote in the bill and the "Present" option means "secret vote". The "Nay" and "Yea" options are at the highest absolute values. We use the results of the roll-call vote of the Brazilian Chamber of Deputies for a period running from 2003 to 2018. This period encloses four complete Legislatures, from the $52^{th}$ to the $55^{th}$. The $52^{th}$ (2003-2006) and $53^{th}$ Legislatures (2007-2010) encompass the first and second term of the former President Luis Inácio Lula da Silva, while $54^{th}$ (2011-2014) Legislature corresponds to the first term of the former President Dilma Rousseff. The $55^{th}$ (2015-2018) started with Dilma Rousseff's second term, but she was forced to leave office in May 2016 and was finally impeached by the end of August 2016. The vice President Michel Temer became the acting President from May to August 2016 and formal President since September 2016 to the end of the term by December 2018. Our analysis reveals signs of political instabilities in the period of 2013 to 2015.

The roll-call vote sequence in a given year of each Federal Deputy in the Chamber provides an unambiguous signature of her/his political behavior. Moreover, by correlating the sequences of each pair of Deputies, we can objectively quantify the level of antagonism and partisanship between them. From this calculation, it is then possible to construct the network of Deputies composing the Federal Chamber. We can build a complete undirected network by connecting every pair of nodes corresponding to Deputies with a correlation distance.

Pearson's correlation coefficient between their corresponding roll-call vote sequences is given by

$$C(A, B) = \frac{cov(A, B)}{\sigma(A)\sigma(B)}, \tag{2}$$

where $cov(A, B) = E[AB] - E[A]E[B]$ is the covariance among $A$ and $B$ roll-call votes sequences, $E$ means expectation value [37], $\sigma(A)$ and $\sigma(B)$ are the variance of $A$ and $B$, respectively. Although, Pearson correlation coefficient possesses interesting properties, such as invariance to linear transformation and limitation to [−1, +1] range, it does not follow the three axioms of a mathematical distance. However, the correlation distance matrix given by $d(A, B) = \sqrt{2(1 - C(A, B))}$ does obey all the distance axioms, and shall be used instead of the correlation as a metric measure of dissimilarity [20]. Correlation distance varies in the [0, 2] range with $d(A, B) = 0$ for $C(A, B) = +1$, all votes are the same, $d(A, B) = \sqrt{2}$ for $C(A, B) = 0$ independent votes, and $d(A, B) = 2$ for $C(A, B) = −1$, totally opposed votes. Closer correlation distance means highly correlated Representatives. Deputies with small distances show similar roll-call vote sequences, while Deputies with generally opposing votes are far apart from each other. The occurrence of highly negative correlations is associated with the presence of two or more antagonist clusters, each one of themselves constituted of partisan Deputies.

We can extract more information direct from the correlation coefficient to show the evolution of polarization degree with time. Fig 1 shows the distributions of the correlation

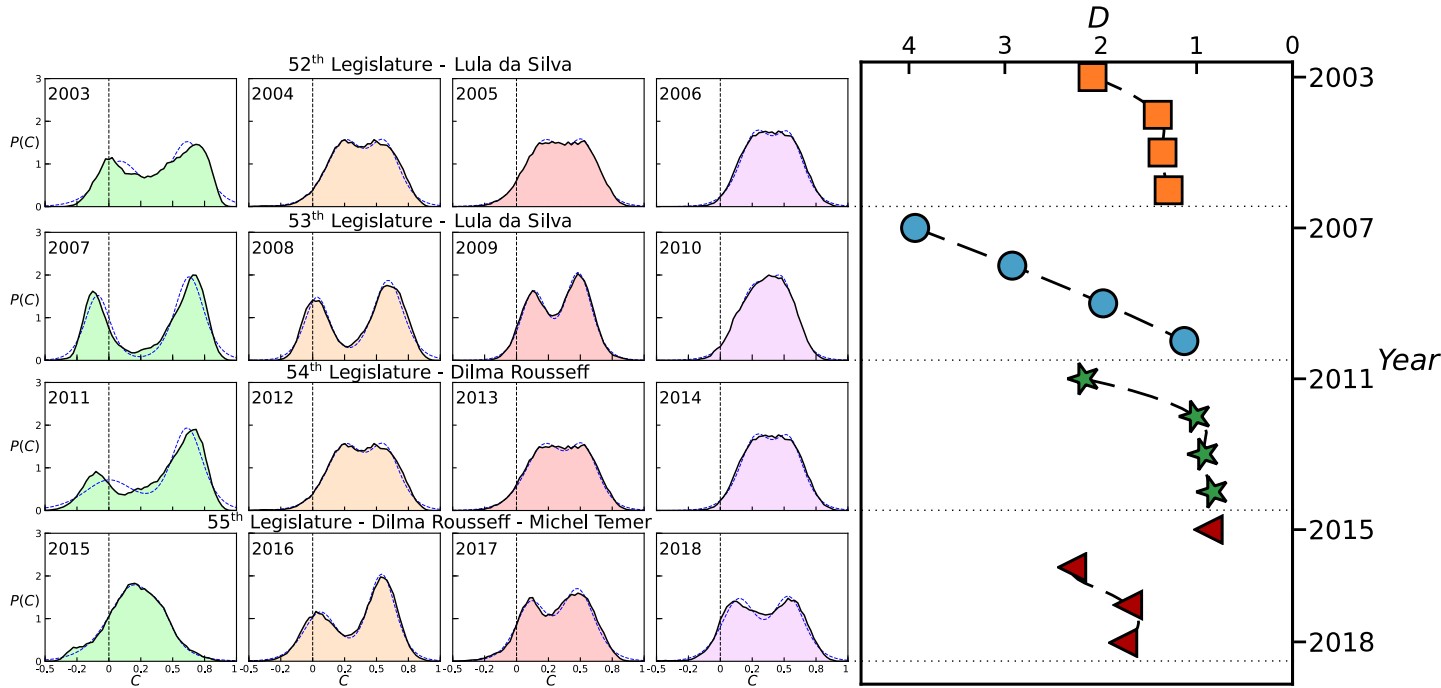

**Fig 1. (Left)** Probability density distributions of the correlation coefficients obtained from each pair of roll-call vote sequences of Brazilian Federal Deputies during the 52[th] (top), 53[th] (center-up), 54[th] (center-down) and 55[th] (bottom) Presidential Legislatures of Brazil. The blue dashed lines are the best fits to the data sets using the bimodal logistic distribution, Eq (3). In all cases, the coefficient of determination follows $R^2 > 0.99$. For the first three Legislatures, the distributions clearly evolve over time, within each legislature, from a bimodal to a unimodal shape. In the 53[th] legislature, this transition occurs between the final years (2009 and 2010). In the case of the 52[th] and 54[th] legislative terms, however, it already takes place between the initial two years (2007- 2008 for 52[th] and 2011—2012 for 54[th]). In contrast, in the 55[th] legislative term, the first year begins with a unimodal shape (2015) with the bimodal shape appearing only in the second year (2016) remaining until the end of the term. **(Right)** Time evolution of the bimodality index computed for the 52[th] (orange squares), 53[th] (blue circles), 54[th] (green stars) and 55[th] (red triangles) Legislatures. The dashed lines are guides to the eye.

coefficients obtained from each pair of Brazilian Federal Deputies for each year during the 4 Legislatures. It is important to note that, within each of the first three legislative terms, the distribution evolves over time from a typical bimodal shape to a unimodal one. A bimodal distribution of correlations with positive and negatives values means that there are groups of Deputies in opposition to each other. The area of negative correlations, meaning strong opposition among Deputies, can be a substantial fraction (more than 17%, 44% and 25% in the years 2003, 2007 and 2011) in the beginning of the term, while the positive correlation area can reach more than 97% of all values in the final year of the terms (years 2006, 2010 and 2014). During the $53^{th}$ Legislature (second mandate of Lula da Silva), this transition only took place effectively between the final years (2009 and 2010). In the case of the $52^{th}$ and $54^{th}$ Legislature (first mandates of da Silva and Rousseff), however, it occurs between the two initial years (2003—2004 for da Silva and 2011- 2012 for Rousseff). In contrast to the previous first three years of the $52^{th}$, $53^{th}$ and $54^{th}$ legislatures (see the distributions for 2003, 2007 and 2011 in Fig 1), the first year of the $55^{th}$ Legislature present a is much less evident bimodal shape. At the end of 2015 a lawsuit was filed to prevent Dilma Rousseff's continuing to act as president. The process began on December $2^{nd}$, 2015 and ended with the impeachment of President Dilma Rousseff on August $31^{st}$, 2016 when vice President Michel Temer became the acting President. The bimodal shape reappeared in 2016 (see Fig 1) with more than a 16% fraction of negative correlation values and the bimodal behaviour was maintained until the end of the $55^{th}$ Legislature.

In order to better quantify the evolution in time of the shape of the distribution of correlations, we created a bimodality index $D \equiv |\mu_1 - \mu_2|/\sqrt{\sigma_1^2 + \sigma_2^2}$ by fitting the data to a bimodal distribution, finding the peak positions $\mu_1$ and $\mu_2$ and their respective standard deviations $\sigma_1$ and $\sigma_2$. The fitting procedure was performed on the cumulative probability distribution (CDF) correlation data obtained after sorting the numbers from the smallest to the largest in a Zipf-Plot [38]. The probability density function (PDF) was obtained by the derivative of the CDF. We observed a best fitting with two logistic distributions given by:

$$P(x; \mu_1, \sigma_1, \mu_2, \sigma_2, b) = \frac{b}{4\sigma_1} \mathrm{sech}^2\left(\frac{x-\mu_1}{2\sigma_1}\right) + \frac{(1-b)}{4\sigma_2} \mathrm{sech}^2\left(\frac{x-\mu_2}{2\sigma_2}\right), \qquad (3)$$

to obtain the fitting parameters $\mu_1$, $\sigma_1$, $\mu_2$, $\sigma_2$ and $b$. Accordingly, a probability density distribution is considered to be bimodal if $D > 1$ [39]. As depicted on the right part of Fig 1, for the first three legislative terms there is a usual behaviour of beginning with a strong dissension evolving into a consensus by the end of the term. Observe that $D$ index for the $53^{th}$ decays in a practically linear fashion, showing that the second term of Lula's Presidency was the most polarized term because it evolved into a consensus only at the end of the term. On the contrary, the $55^{th}$ legislative term begins with the lowest value of bimodality ($D = 0.8762$) in 2015 (namely, the year of lawsuit) but the bimodality increases again in the second year of the term in 2016 (namely the year of impeachment) and decreases as the Legislature goes to the end.

At this point, we show that the bimodal shapes, as well as the presence of a relatively large fraction of negative values in the distribution of correlations, are closely associated with the polarization of the votes in the Chamber of Deputies. Now we can proceed to a deeper analysis of the time evolution of the Representative dynamics applying network analysis tools, such as Minimal Spanning Trees (MST) of correlation distances among Deputies. The visualization of the same correlation data after a proper rearrangement provides much more information about the players and events in a given year.

MST is a network that connects all the vertices without any loops with the smallest distance pathway [40]. It is important to highlight that MST is completely blind to any information that

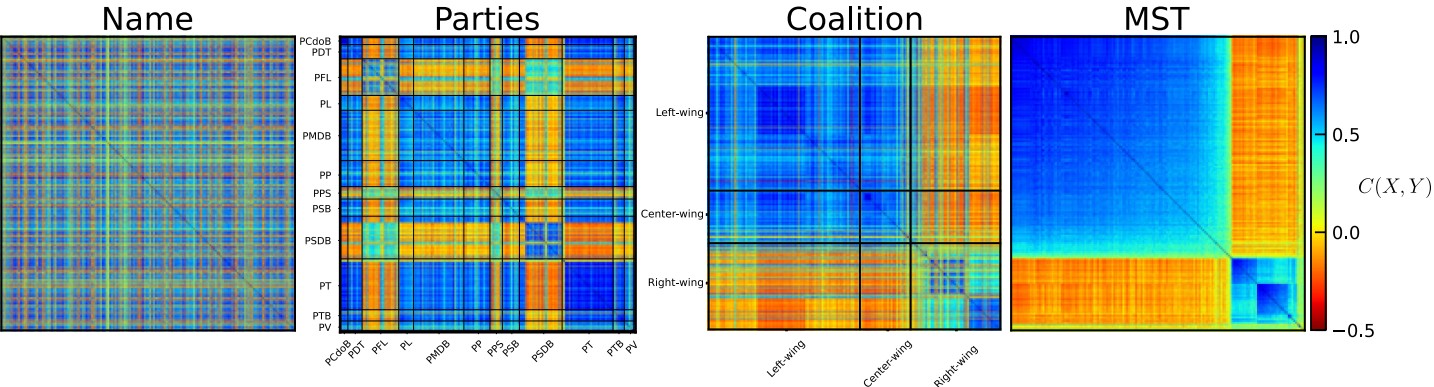

**Fig 2. Different ordering processes imposed over the Deputies correlation matrix for 2007.** The matrix on the **left** shows the alphabetical ordering by name; the **center-left** figure shows the ordering by political party. For better visualization, the solid black lines delimit the parties with more than 10 Deputies. In the **center-right** we found the political coalition ordered correlation matrix. In this case the solid black lines delimit the three possible wings for the Federal Chamber (Left-wing, center-wing and right-wing). Finally, on the **right** the MST-Prim ordered correlation matrix is exposed. Observe how the MST-Prim ordering shows defined cluster without any political information.

influences the votes such as party, seeing only the revealed correlation distance between the Representative votes. There are two main algorithms to obtain the MST, Prim [41] and Kruskal [42], with the same final draw of the network. Although Kruskal's algorithm is faster, Prim's algorithm evolves by the closest neighbors sub-networks, while Kruskal draws all sub-networks in parallel. Prim, therefore, is already organizing the network using the strongest connected neighbors, and is ideal to re-order the correlation, or correlation distance, matrices. We MST-prim re-ordered the correlation matrices, to the best of our knowledge, by the first time. The re-ordered matrices shall display the cluster of closer Deputies.

Fig 2 shows the correlation matrix for the year 2007 arranged by the name (which should represent a random matrix), the political party, the political coalition, and the MST-Prim re-ordering of Deputies. We can see a close resemblance of the MST-prim with the coalition matrices. It is worth noting that MST-prim reordering does not need any extra information about the party and/or coalition of each Deputy. Moreover, the two arrangements could be completely different in a situation which is completely absent of partisan fidelity, with the clusters formed of other similar characteristics.

Fig 3 shows the correlation matrices organized accordingly to the MST-prim network draw for three outstanding years (2007, first year of Lula's second mandate; 2015, first year of Rousseff's second mandate and 2016, first year of Temer's mandate). In the center we present the Representative MST network draw with the vertices' color representing the Deputy party. On the right, we expose the same reordered MST correlation matrix but in this case, each Deputy's color is given according to party, instead of the correlation value. In 2007, the MST reordered matrix clearly shows opposing clusters (two main clusters of the MST network). There are two large clusters representing the government (the large block in the upper-left) and the opposition (the smaller block in bottom-right), but we can visualize sub-blocks in the opposition cluster as well. In the party-colored matrix on the right we can see that the large upper left cluster, the government block, composed mainly of the red (PT) and blue (PMDB) parties. In this block there is a continuous transition between hard supporters at the far upper-left to light supporters at the bottom-right boundary. The opposition lower right cluster is composed mainly of the yellow (PFL) and green (PSDB) parties, two well cohesive Representatives. Although, these two opposition clusters tend to agree with each other, there is a clear division between them. The MST network also shows these features, especially in the yellow and green

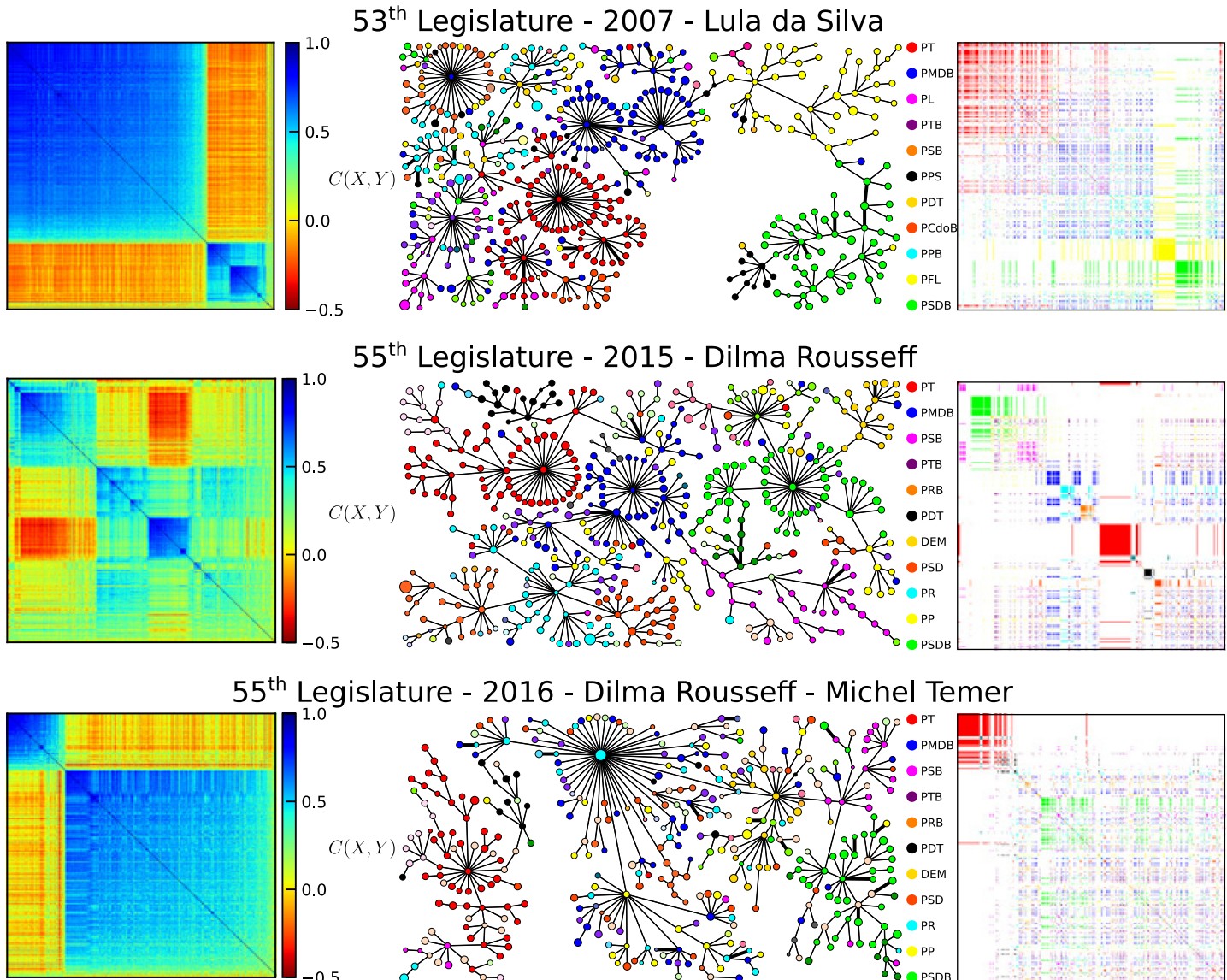

**Fig 3. On the left are the correlation matrices ordered by the MST network for 2007, 2015 and 2016.** The colours correspond to the values of the Pearson's correlation coefficient between pairs of Deputies, calculated from their roll-call vote sequences in the respective year. The positions of each Deputy are ordered according to the sequence of construction of the corresponding MST, as explained in the text. In the **centre** are the minimal spanning trees (MST) extracted from the network of correlations among Deputies for the same years, computed using Prim's algorithm [41]. In these networks, the colour cluster of vertices were imposed according to the Deputy political party and vertice's sizes are associated to the number of votes in which Deputies were elected. The correlation matrices maintaining the same ordering are shown again on the **right**, but now, instead of correlation coefficient value colours, the colours of each point in the matrix correspond to the Deputy's political party.

sub-networks spatially segregated. In the large government cluster one can observe a tightly connected red block and more dispersed sub-groups, explaining the blue gradient of the large cluster in the left matrix. By looking at the party colored matrix it is clear that there is also a red-blue segregation, with red mostly contained in the upper left corner while the blue is contained at the center and borders of the large government cluster. That means PMDB is associated with the government with much looser ties than PT itself.

In 2015 the color pattern changes abruptly showing the absence of a strong polarization, and several clusters. Only red color (PT) is clearly segregated in the central cluster, with strong

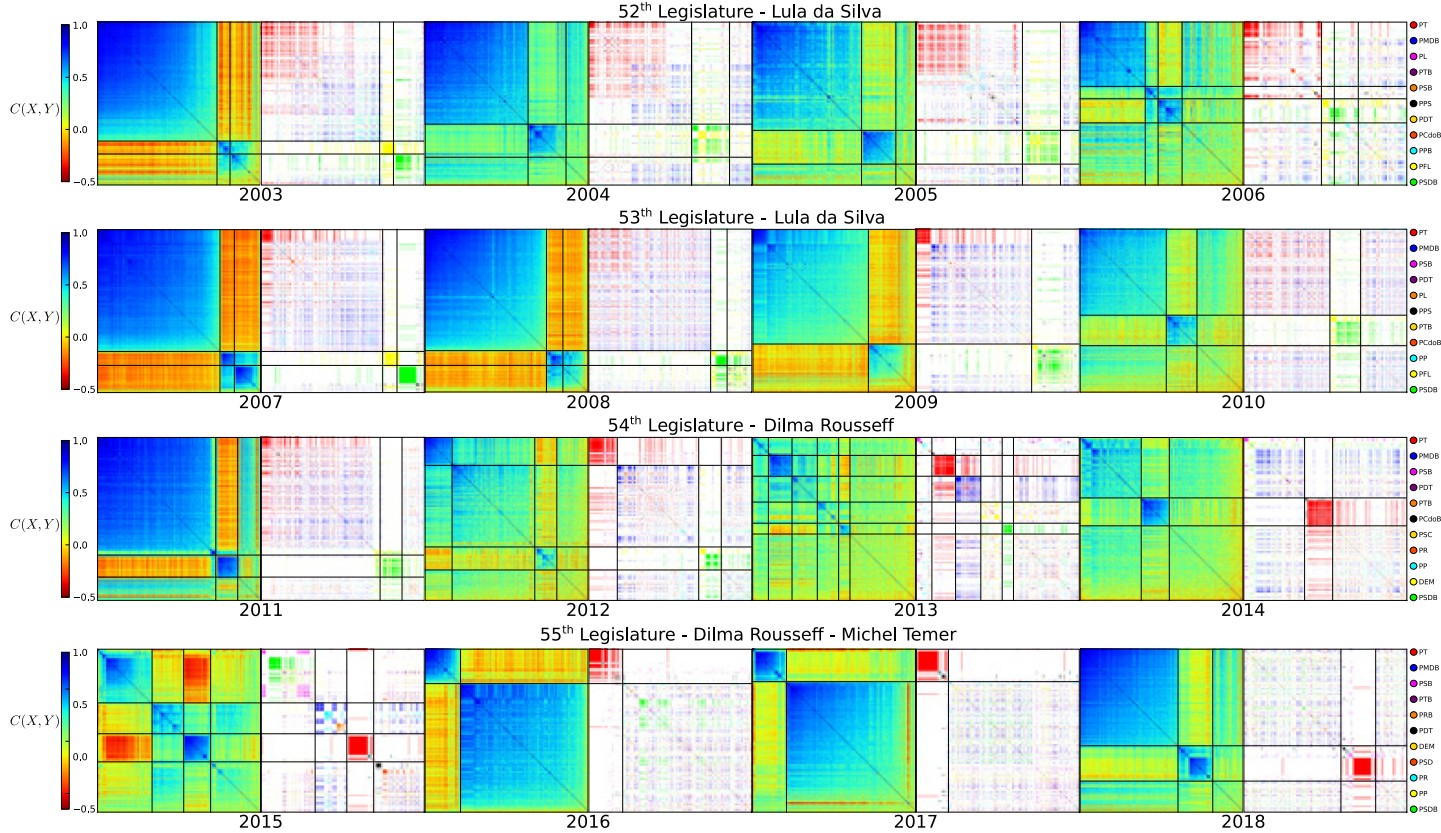

**Fig 4. MST reordered correlation and party matrices for the 52th, 53th, 54th and 55th Legislatures.** In each panel, in the **left** we present the MST-prim reordered correlation matrices colored by the correlation values, and on the **right** the same matrix colored by the party. We maintain the same color for the parties throughout all Legislatures. The MST ordering process can be used to detect strong anomalies in the Representatives' system.

anti-correlations with the upper left cluster. The green party is still cohesive but more spread than before. This could be explained by the gradual lack of political support of the Chamber to the Government, which eventually led to the impeachment of President Dilma Rousseff in 2016. In fact, the government block (blue and red parties in 2007) in 2015 got isolated and a strong negative correlation is observed between the remaining government party (red party) and the opposition block (green party in 2015). The strong polarization against the remaining government party is clearly observed in the reordered matrix for 2016. The small left up cluster corresponds to the remaining government party. These pictures show a large government support cluster breaking down and turning upside down with the government cluster isolated due to a lack of supporters.

In this stage, we notice by comparing the three outstanding years (2007, 2015 and 2016) that the MST network obtained is capable of detecting political instabilities in the Representatives systems. A more detailed view of the political instability can be observed by inspecting all reordered correlation matrices. Fig 4 shows the reordered MST correlation matrices together with the party MST ordering, side by side, for 2003 to 2018. A thin black line was drawn for each visible cluster from the MST-prim ordered matrix and extended to the party colored matrix on the right. In the 52th and 53th legislature, despite the transition from the polarized to consensual state, the pro government cluster remains from the beginning to the end of both legislatures. The government coalition PT/PMDB forming a large cluster while PSDB/PFL (DEM in the 53th legislature) forming the opposition cluster. As the polarization degree

diminishes, the cluster becomes more dispersed. However, from the second year of the 54th (2012) the main pro government party (red party) began to isolate from its coalition political parties, with a clear segregation observed in the correlation colored matrix, instead of the just color gradient observed before, correspondingly to red (PT) and blue (PMDB) parties. This evolved to the isolation of the red group by the end of the term in 2014. The 2013 year also shows an unusual splitting pattern in few strong correlated groups and large uncorrelated dispersed groups. This year corresponds to the strong protests that started with bus ticket price movements [43, 44]. The first year of Rousseff's second term (2015) shows a strong government isolation and ruptures in several clusters. The red cluster isolation remained until the end of the 55th legislature (2018). The impeachment process was launched by the end of 2015, and ended by the mid of 2016. In 2016 the usual pattern of two clusters, a large one and a smaller one, reappeared. Only then, the small one was comprised almost solely by the red (PT) while the large cluster was dispersed through all the other parties, with almost no color segregation, especially in 2017 and 2018, with a small green segregation in 2016. This clearly shows that the once strong government PT became the well connected but small opposition. It seems like a party under attack trying to defend itself, because the party cohesion is much stronger than in the years it was in charge of the country.

The null model of our data was obtained by shuffling the votes among the Representatives and bills, keeping the same distribution for the votes. The result is shown in the supplemental material. The MST-ordered correlation matrices did not show any special feature nor cluster, the MST network do not show any color segregation. This shows the cluster and regularities we observed cannot be explained as random events, but rather as an emergent behavior of the Representative's decisions.

A summary of our observations is: **(1)** MST-Prim reordering of correlation distance matrices of a complete network is capable to display the inner clusters of Representative's similarities. **(2)** There are more than two clusters of Representatives showing clearly different degrees of disagreement/agreement between them, with two main strong clusters, regardless of the Deputy's party. **(3)** The correlation matrices shows between three to five visible clusters, which can be interpreted that a number from 5-6 parties would be more than enough to capture all ideological/philosophical positions of the whole society through their Representatives. Although it also shows that only two parties are not enough to capture the whole diversity of thinking of Brazilian Representatives, the present number of more than 30 parties is not necessary. **(4)** The degree of disagreement is stronger in the beginning of a Legislature and evolves to a consensus by the end of the term. **(5)** Disassemblement of the large government cluster is a sign of political instability, such as the one starting in 2012 evolving to strong government isolation in 2014. Total break up of this cluster, such as the one that happened in 2015, is a strong sign of a political crisis. The former party in charge (PT) became completely isolated after 2016, but more cohesively bound.

## Discussion

Here we want to emphasize the following aspects of our methodology: **(1)** It is blind to any bias seeing only the revealed final result of the members behaviour. Any regularity observed can then be further and analysed in a greater depth, to search for explanations for the findings; **(2)** It shows connections between the members, regardless of the topic of the bills. Any analyses of the bill contents would require a step further into the semantics of the bills; **(3)** All data was publicly available on the internet, as required by Brazilian transparency laws; **(4)** All data refers to Deputy's real votes, and not answers to hypothetical questions. We could say we only observed revealed Deputy's preferences; **(5)** We observed that, in periods of political stability,

presidential terms begin with a strong polarization evolving to a consensus by the end of the term. This pattern disappeared in the presence of political instability. **(6)** We observed that 5 to 6 parties would be enough to include all ideological positions in Brazil, meaning that the more than 30 existing parties is a Brazilian idiosyncrasy that needs further comprehension; **(7)** We observed the onset and the peak of a political instability that led to a presidential impeachment, characterized by the breakup of a strong government support cluster and posterior isolation of the former party in charge; **(8)** the MST-prim reordered correlation matrix was capable of discriminating the different clusters without any further information than the publicly available votes. The comparison of side by side matrices colored by correlation value and parties shows that the clusters observed in this rearrangement carries a handful of information.

Although it is questionable to establish definitive hypothesis to explain our results based only on this set of data, we can at least raise some possible explanations. The possible explanations about the dissension/consensus time evolution are: First, using the Machiavellian hypothesis, a new government tends to take unpopular measures in the beginning of their mandate and the most popular ones by the end of the term, especially when re-election is allowed. Another hypothesis is that governments tend to get Representatives support with time, bringing most of them to their side. Game theory [45], however, shows that cooperation emerges by reciprocity in repeated games. Governments tend to learn what can be approved by that group of Deputies, which, on their side, also tend to learn what can be possible with that government. This learning curve should screen the proposed bills to only ones with better chance of approval. This would mean a large consensual approval of the bills in the last periods of the term. Another fact to be taken into account is that there are, also, bills being rephrased in brand new bills in order to assure their approval, with the consequent rejection of the original bills. This would mean not only a consensus for the approval of new bills, but also an agreement for the rejection of the original bills. The question raised if the observed consensus building during a Legislature happens to better serve the representative voters or is a betrayal to the voters, cannot be answered by our content blind methodology. The only thing our technique can do is to highlight clusters of Representatives and bills with regularities deserving a deeper understanding.

There are also a number of hypotheses to explain the large unnecessary number of parties in Brazil (32). The fact that there are a large number of parties represented in Congress means that a large number of Deputies received enough votes to be elected even when they did not belong to the 3 or 4 main parties. The first hypothesis is that people created parties "for rental" to later negotiate with the bigger main parties. The question is, then, how they managed to get enough votes to exist as a party. Another hypothesis, is that Brazilians tend to vote in the candidates they know well, regardless of their party and or ideology, which facilitates the existence of large numbers of parties. The polarization of the Representative's votes in Congress observed in this report, however, shows that Representatives tend to have a firm position, especially in the beginning of the term. That means that there are some commitment to the population's votes with political position, or that, there is a pre-selection of candidates that have more firmer political position. Another possibility for the large number of parties lies in the party's directions. If there is no internal democracy inside the parties, and just a small number of leaders who decide who will run as a candidate or not, then candidates that see themselves as competitive tend to create another party, or join smaller parties, to be able to run for the position.

Our observation also shows that 2015 was a completely atypical year, not only when compared to the past first years of a presidential term, but in all aspects. The presidential election was practically a tie (51.64% vs. 48.36% of valid votes) [46] and an economic downturn fueled protests in the whole country. Although, Rousseff's impeachment happened only in the middle of 2016, the process started in 2015 and generated "noise" during the whole year. The "Car

Wash" operation, which started in 2014 and reached the first politicians in 2015, also generated political unrest in the country. MST ordered correlation matrix shows the large cluster of government supporters observed in previous years vanishing, and the hard core of presidential supporters becoming isolated in Congress. We can also observe that the onset of the government support group started in 2012, by the splitting of two groups (red and blue) that were completely mixed a year before. In 2013, the year when protests exploded, the red and blue groups segregation became stronger with the two parties still weakly correlated, finally evolving to anti-correlation in 2015 with strong isolation of the red (PT) party in charge. Therefore, the MST-prim reordering of the correlation matrix can provide an indicator of political stability.

In conclusion, our approach, based only on blind observation of the revealed votes in Federal Chamber of Brazil, proved to be a powerful method to analyze the socio-political behavior of its Representatives. It enabled us to investigate how the consensus/dissension evolves with time during a legislature term. It also pointed out political instabilities as shown in 2013-2015 data. This analysis opens up a number of questions that could then be studied by a deeper analysis of the bills proposed and the time evolution of the relationship between Government and Representatives. We also believe that the same methodology could show similar results if applied to other countries or organizations.

## Supporting information

**S1 File. Supporting information.** Data mining and manipulation of roll-call votes sequences, Minimal Spanning Tree computation, correlation distribution fitting parameters and null model.
(PDF)

## Acknowledgments

We gratefully acknowledge the Brazilian agencies CNPq, CAPES and FUNCAP, and the National Institute of Science and Technology for Complex Systems in Brazil for financial support. L.M acknowledges David Quispe Aruquipa for the help provided with the data mining processing and Biviana Orbegozo for her valuable comments that improved the manuscript. C.L.C acknowledges CNPq grant (312049/2014-5); FAPESP (11/51959-0); "Física do Petróleo em Meios Porosos", PETROBRAS-UFC 2016/00328-4 and WIPPS II—Simulação numérica de invasão de água em poços produtores", PETROGAL-UFC.

## Author Contributions

**Conceptualization:** Ludwing Marenco, Carlos Lenz Cesar.

**Data curation:** Ludwing Marenco.

**Formal analysis:** Ludwing Marenco, Humberto A. Carmona, Felipe Maciel Cardoso, José S. Andrade, Jr., Carlos Lenz Cesar.

**Investigation:** Ludwing Marenco.

**Writing – original draft:** Ludwing Marenco, José S. Andrade, Jr., Carlos Lenz Cesar.

**Writing – review & editing:** Ludwing Marenco, José S. Andrade, Jr., Carlos Lenz Cesar.

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
