## [Decision Letter · Decision Letter 0]

2 Dec 2019

Time evolution of Brazilian legislative Representatives` behaviour using a complex networks approach

PONE-D-19-31085

Dear Dr. Marenco,

We are pleased to inform you that your manuscript has been judged scientifically suitable for publication and will be formally accepted for publication once it complies with all outstanding technical requirements.

With kind regards,

Haroldo V. Ribeiro

Academic Editor

PLOS ONE

1. We suggest you thoroughly copyedit your manuscript for language usage, spelling, and grammar. If you do not know anyone who can help you do this, you may wish to consider employing a professional scientific editing service.  

"No the funders had no role in study design, data collection and analysis, decision to publish, or preparation of the manuscript.".

Please provide an amended Funding Statement that declares *all* the funding or sources of support received during this specific study (whether external or internal to your organization) as detailed online in our guide for authors at http://journals.plos.org/plosone/s/submit-now.  

Please state what role the funders took in the study.  If any authors received a salary from any of your funders, please state which authors and which funder. If the funders had no role, please state: "The funders had no role in study design, data collection and analysis, decision to publish, or preparation of the manuscript."

* Please send your amended statements by return email; we will change the online submission form on your behalf.

Reviewers' comments:

Reviewer's Responses to Questions

**Comments to the Author**

1. Is the manuscript technically sound, and do the data support the conclusions?

Reviewer #1: Yes

Reviewer #2: Yes

2. Has the statistical analysis been performed appropriately and rigorously? 

Reviewer #1: Yes

Reviewer #2: N/A

3. Have the authors made all data underlying the findings in their manuscript fully available?

Reviewer #1: Yes

Reviewer #2: Yes

4. Is the manuscript presented in an intelligible fashion and written in standard English?

Reviewer #1: Yes

Reviewer #2: Yes

5. Review Comments to the Author

Reviewer #1: The use of advanced statistical methods and data science to study opinion and vote dynamics has been the focus of many research efforts over the last decade. However, most previous studies have focused on the collective dynamics before elections. Somehow differently, in this paper, the authors follow the interaction among Representatives once in office. The collected and analyzed a large dataset consisting of all votes of the 500+ member of the Brazilian Federal Chamber over four presidential terms (16-year period).

Based on the time series of the vote of each member, they compute the correlation between members. From that, they constructed a complete undirected network, where the weight of the link between each pair of nodes (member of the Chamber) is given by the correlation (distance) between their time series. They then obtained the Minimal Spanning Tree (MST) and regrouped the nodes based on their relative positions (and distances) in the MST. With this novel approach, they managed to characterize the clustering dynamics of the members of the Chamber. In particular, they identified a handful of clusters that evolve in time. Somehow surprising, they show that the degree of disagreement among groups is strong in the beginning, but it evolves towards a consensus over time.

During the period under study, there was a successful presidential impeachment in Brazil. What this study shows is that the political instability is preceded by a shrinking of the cluster of members usually supporting the government. This result suggests that the dynamics of the cluster of members supporting the government might be considered as a predictor of future political instabilities.

The paper is definitely interesting, and the results are scientifically sound. Presidential impeachments is a hot and timely topic and it will very likely attract the attention of a broad audience. Besides, the novel approach reported here can in fact be applied to many other systems where the dynamics of correlations are mapped into a time-dependent network. I am sure that future studies will consider applying it to other systems. Since the paper is well written and the description of the methods and approach is clear, I recommend publication in the present form.

Reviewer #2: Report on the manuscript PONE-D-19-31085

Time evolution of Brazilian legislative Representatives’ behaviour using a complex networks approach

by L.Marenco et al.

The authors study the emergence of communities and other features in complex networks that were assembled by using the data of actual votes delivered by the deputies in the Brazilian house of Representatives during 16 years. This is a relevant theme within the context of social physics.

For each year a network is obtained, where the nodes represent the deputies, and weighted connections between each pair of nodes are obtained by first evaluating the correlation between similar votes for all bills that were subject of decision in that year. The correlation matrix is subsequently transformed into the correlation distance, which is the source of the connection weights. The authors present two kinds of results: the first one is the distribution of the values of the correlation matrix elements, which switches between the bimodal and single bell shapes; the second one is the community structure of the correlation matrix, which was achieved by the evaluation of the minimal spanning tree for each network. This evaluation was carried out with the help of the correlation distance matrix. The results first indicate a dynamical evolution in the bimodal/unimodal character of the correlation distribution, with some regularity for the first three legislatures. A leave of this pattern was observed for the fourth legislature, which was marked by a political crisis that resulted in the presidential impeachment. The second series of results indicate that, without previous knowledge of the party affiliation or inclusion in the majority/minority groups supporting the ruling president, minimal spanning trees are able to uncover clusters of nodes within the network that keep strong relations with the known distribution in these groups. The clustering results only from the knowledge of actual votes.

The methodological aspects of the work are clearly stated, as well as the used data. The conclusions emphasize that the results do not require any previous knowledge of the political affiliations of the deputies, and that the used methodology can be used to obtain insights on the support of a president in the house of Representatives.

Previous studies have addressed the use of network methods to analyze political forces in Representative houses in other countries, as correctly acknowledged by the authors. However, I understand that the current work adds new insights into the field. Therefore, I support the publication of the manuscript in Plos One.

6. PLOS authors have the option to publish the peer review history of their article (what does this mean?). If published, this will include your full peer review and any attached files.

Reviewer #1: No

Reviewer #2: No

---

## [Editor Report · Acceptance letter]

8 Jan 2020

PONE-D-19-31085 

Time evolution of the behaviour of Brazilian legislative Representatives using a complex network approach 

Dear Dr. Marenco:

I am pleased to inform you that your manuscript has been deemed suitable for publication in PLOS ONE. Congratulations! Your manuscript is now with our production department. 

With kind regards,

on behalf of

Dr. Haroldo V. Ribeiro 

Academic Editor

PLOS ONE